# Does Volunteer Service Foster Education for A Sustainable Future?—Empirical Evidence from Chinese University Students

**Ling Chen [1], Degang Li [2],* and Yong Li [3],***

1 The Communist Youth League Committee, University of International Business and Economics, Beijing 100029, China; chenling@uibe.edu.cn
2 School of Economics, Beijing International Studies University, Beijing 100024, China
3 School of Law, University of International Business and Economics, Beijing 100029, China
* Correspondence: lidegang666@126.com (D.L.); 01680@uibe.edu.cn (Y.L.)

**Abstract:** Sustainable development is an extremely important global vision of the current era, and higher education plays a significant role in deeply immersing young people in this prospect. As an important part of labor education for contemporary Chinese university students, volunteer service has proven to be a sustainable and valuable talent training process. Different from the grand narrative and simple political perspective of previous studies, this paper selects the relevant data of a university in Beijing as a sample to conduct a "black box disassembling" micro study. By building three sets of data, the authors were able to adopt difference-in-differences, analysis of variance, and hierarchical regression methods to study the comprehensive improvement of university students' inner ability and sense of responsibility after participating in voluntary services. Results show that participation in more voluntary services will improve students' performance and employment quality, mostly promote students' teamwork ability, and differentially enhance students' comprehensive, personal, and collective senses of responsibility, thus facilitating education for a sustainable future and, accordingly, economic development. The study reveals the formation mechanism of the sustainability value of voluntary service in more detail, complements the absence of Chinese sample data in international studies, and lays an effective foundation for more comprehensive micro studies in the future.

**Keywords:** volunteer service; Chinese university students; sustainable future; education evaluation; quantitative research

## 1. Introduction

The far-reaching significance of sustainable development for today's world has been emphasized in recent years. In 2015, the United Nations adopted the 2030 Agenda for Sustainable Development, which aims to promote prosperity for people and the planet, strengthen peace and freedom in the world, eradicate poverty in all its forms and manifestations, and achieve sustainable development. The Agenda aims to achieve its goals through 17 specific Sustainable Development Goals and a 15-year plan vision, which further become a list of actions for people and the planet and a blueprint for success [1]. In order to achieve the goal of sustainability, the leading factor of "people" plays a decisive role, and both individual and collective efforts around the goal can bring us closer to real sustainable development outcomes. Human development and growth cannot be separated from the education system, and sustainability as a mode of thinking or behavior process is mainly obtained through the training of people in the education system. In 2017, UNESCO published Education for Sustainable Development Goals: Learning Objectives [2]. The report argues that education is key to achieving sustainable development by developing relevant knowledge, skills, values, and attitudes. Education for Sustainable Development (ESD) aims to develop the capacity of individuals to reflect on their actions and to consider

their current and future social, cultural, economic, and environmental impacts from local and global perspectives. Through ESD, individuals make decisions in complex contexts in a sustainable way, enabling their own actions and those of society as a whole to evolve towards sustainable development. In Education for Sustainable Development: Learning Goals, key competencies for sustainable development are clearly addressed: systematic thinking ability, anticipation ability, normative ability, strategic ability, cooperative ability, critical thinking ability, self-awareness ability, and comprehensive problem solving ability. These abilities are not entirely dependent on students' knowledge-learning activities in the classroom, but also need to be acquired through appropriate scenarios and structured and standardized training as far as possible. As a service-oriented learning model that has attracted much attention in recent years, volunteerism has occupied a place in sustainable development. Its powerful value is mainly reflected in its impact on the growth process of college students participating in it as an educational approach. In early studies, foreign university students' participation in volunteer service was found to have an impact on their academic performance, concepts of values, personal abilities, career choices, etc., which reflects the substantive value of participation in volunteer service and is also the cornerstone of this paper.

Volunteering has fully entered the campus life of Chinese college students since the 2008 Summer Olympic Games in Beijing, which showed the world a new perspective on Chinese youth. In 2020, the CPC Central Committee and the State Council issued and implemented the Opinions on Comprehensively Strengthening Labor Education in Universities, Primary and Secondary Schools in the New Era, emphasizing the importance of voluntary service as a part of labor education. Different from social administrators who focus on the realization of volunteer service, researchers pay more attention to the value of volunteer service to the growth of university students. Volunteer service builds a vivid environment for Chinese students to learn and grow up in outside the classroom, and is an important bridge connecting university students with society and the state. Accordingly, it becomes necessary to explore the internal mechanism of volunteer service in promoting Chinese students' growth.

Studies on Chinese university students' voluntary service in international journals have paid more attention to the political attributes of this behavior, and have found that the country has shaped university students' behavior as model citizens and held firm to the ideals and beliefs of the country by means of voluntary service [3]. It was believed that volunteering was utilized for strengthening the state's ideological hegemony, implementing innovative social management for social stability, and facilitating the Chinese Communist Party's building for its long-term rule [4]. It was also considered that volunteerism had evolved into an area where the government controlled and managed the emerging civil society, and that China's officially led volunteerism programs were highly formalistic and inductive, and could not satisfy students' strong desire for meaning and value in volunteerism compared with bottom-up volunteering [5]. Such a research perspective has certain value, reflecting the overseas academic circle's long-term attention to the field of Chinese ideology, but it cannot fully reflect the role of voluntary service on individual Chinese university students. It is of obvious practical value to study whether Chinese university students have changed their academic performance, personal values, and personal abilities after participating in voluntary service, which also will provide a complete evaluation of the sustainability value of this educational approach.

In the past, without the help of information systems, Chinese researchers used questionnaires and other methods to collect data on voluntary service, which permitted a certain amount of memory bias on the part of the respondents [3]. Moreover, due to the limited coverage of questionnaires, only sampling studies could be conducted in most cases. With the help of a new information system, it is possible to obtain more accurate and true full-sample service records. Combined with other firsthand data, the scope of research has been greatly expanded, breaking through the limitation of mainly investigating the subjective feelings of university students, and realizing systematic data analysis. Therefore, based

on the limitations of the existing research and combined with the new reality, this paper tries to make the following innovations: first, part of the data in this study was made up of objective data recorded by the information system for research; secondly, it introduces the difference-in-differences method, which is often used in public policy research; thirdly, three sets of data are used to discuss the sustainable value of the volunteer service of contemporary Chinese university students from three dimensions: the measurable ability result, evaluable inner feelings, and evaluable external responsibility. Due to the diversified requirements of data types, and according to the maximum availability of data, the final sample data were uniformly sourced from a university directly under the Ministry of Education in Beijing, which has typical and well-recorded cases of volunteer service.

## 2. Literature Review

### 2.1. The Value of Volunteer Service in Higher Education

University has always had a strong impact on students. The educational learning environment created by faculty and students, all the experiences in university, and the role of peer groups all together shape students in terms of their tendency to become involved and foster stronger relationships and intellectual abilities, and help them develop more meaningful philosophies of life [6]. University students have much to gain from volunteer service: early studies have confirmed that student volunteer service significantly increases their satisfaction with university community service opportunities and university leadership development, especially regarding the cultivation of student leadership [7]. Recent research has described in more detail how this desired leadership gain manifests itself in volunteers [8].

American universities have formed a rather fixed service-learning mechanism for volunteer service, which originates from the demand for volunteer service modules in higher education, but goes beyond such a demand. Combining students' community service with the academic curriculum, this mechanism requires students to review their own experience through writing, discussion, classroom presentation, and service learning. It enriches the traditional curriculum by giving students the opportunity to test or demonstrate abstract theories in the real world and improve the quality of their service by providing intellectual support [9]. Such volunteer service, conducted in the form of courses or otherwise, is regarded to have a significantly positive impact on students' mentality, awareness, readiness for student service, and desire for higher degrees, and promotes a stronger sense of social responsibility, greater recognition of the value of education, and more interest in multicultural policies and diversity issues [10].

The indexation of those benefits shows that volunteer service has a significantly positive impact on 11 indicators, e.g., academic achievement (*GPA*, writing skills, critical thinking), values (commitment to activism and promotion of racial understanding), self-efficacy, leadership (leadership of activities, self-evaluation of leadership skills, interpersonal skills), and intention to participate in service after graduation [11]. A random-effects study using meta-analysis software found positive effects of service learning on academic, personal, social, and civic identity [12]. Some scholars have also found this effect to be partially due to the experiences gained by students through participation in highly engaging social activities in their spare time [13]. Through this exercise, they can better expand their relationships and understanding of others, and see firsthand how people solve collective problems together [14]. More detailed study through the investigation of 358 university student volunteers and 173 non-volunteers in three universities in Wuhan, China, found that volunteer service improved many basic practical abilities, and volunteers expanded their interpersonal communication abilities, social cognition, environmental adaptation, organization and coordination abilities, teamwork abilities and self-expression abilities; this was all achieved through the process of volunteer service [15]. More recent research has also found that service learning improved students' personal citizenship skills [16], enhanced their ability to solve social problems, improved their social effectiveness [17], and motivated a change in the model of volunteering itself, dynamically, with a holistic agenda

being developed to cover service learning, leadership, and employability [18]. If we focus on the topic of sustainability, it is apparent that as early as 2008, scholars stressed the need to include ESD in both formal and informal curricula. The informal courses mentioned here include volunteering [19]. Undergraduate volunteer service also brings realistic academic performance improvement, social skills improvement, personal happiness, and self-esteem enhancement to students, and thus contributes to the further sustainable development of society [20–22].

In-depth studies have also been conducted on the motivation of university students to participate in volunteer service. Results show that volunteer service is an important way for students to achieve low-risk development in interpersonal relationships, time management, career development, and academic and professional performance. This implies that students may aim for personal goals instead of seeking to improve external conditions [23]. Students who have not yet participated in volunteer service attach more importance to the vocational function of volunteer service [24]. Besides skills, material rewards and subsidies are also important motivations [25]. More purposeful motivations include explicit service-learning classroom requirements [26] and specific comprehensive quality evaluation requirements [27]. An individual interview with 50 university students in the UK demonstrated the coexistence of altruism and employability as service motivations [28]. A questionnaire and interview of students in Hong Kong showed that students who participated in service learning because of their own interests had a higher willingness to continue to participate in follow-up volunteer service projects than those who participated in service learning for external reasons. The external motivation was relatively complex, and the motivation of gaining recognition or identity from others was more likely to enhance the motivation for continuous service than the motivation of mandatory graduation requirements [29]. Another survey of 605 university volunteers at sports events in Singapore showed that value expression, community involvement, personal growth, love of sports, and other factors had a positive impact on continued volunteering, whereas extrinsic rewards had a negative impact on continued service [30].

## 2.2. On the New Method Introduced in the Study

When judging whether a public policy is effective or not, researchers often look forward to overcoming the endogenous problems of economic events and economic policies during causal effect analysis and policy evaluation. Common econometric tools include instrumental variables, breakpoint regression, propensity score matching, and the difference-in-differences method [31].

The difference-in-differences method was introduced into the field of economic research in the 1970s [32]. The first scholars to introduce the method in China were Zhou Li'an and Chen Ye of Peking University [33]. The method has been applied in education research in China mainly in the past decade. Focusing on the financial reform of compulsory education, Sun Zhijun et al. discussed the problem of system stagnancy considered with respect to the increase in financial funds [34]. Chen Lin and Xia Jun discussed the negative impact of a university enrollment expansion policy on technological innovation efficiency [35]. Likewise, Li Jian et al. (2018) discussed the impact of university enrollment expansion and found a negative impact on total factor productivity growth in mainland China [36]. Yu Jingwen et al. found that the reform of provinces' governing counties had a positive impact on the supply of public education [37]. Cui Sheng and Wu Qiuxiang found that serving as student leaders can effectively improve students' working skills, interpersonal relationships, and academic performance [38,39]. Zheng Bingdao and Zhu Hanbin studied the eighth new curriculum reform of high school textbooks and found that students became more supportive of the central government and more willing to join the CPC and work in state-owned sectors [40].

With the rapid economic development in China in recent years and the ensuing reforms and changes, policy effectiveness has become a focus of attention for many parties. In fact, the difference-in-differences method is mainly useful for treating the implementation of

a public policy as a natural experiment. The sample is divided into the treatment group, which is affected by the policy, and the control group, which is not affected by the policy. The variations in the treatment group and the control group are obtained with regard to the first difference: before and after policy implementation. The second difference is evaluated on the basis of the variations between the two groups. The net effect of policy implementation is obtained without being affected by the endogenous problems of the data.

## 3. Research Materials and Methods

The sample university selected in this paper is a medium-sized, double-first-class university in Beijing, which has a population of about 2200 undergraduates per year. Majors of the university cover economics, management, liberal arts, law, science, and engineering, and its students come from 31 provinces and autonomous regions in China, which is representative of the overall sample of universities.

In the first half of 2012, according to the requirements of the Beijing Volunteer Service Federation, all university students were encouraged to register as volunteers on the official website of Volunteer Beijing to apply for and record the time of their volunteer service on and off campus. The sample university has actively implemented this initiative. Starting with the 2012 class enrolled in September of that year, all students were asked to register for and use this system. In 2012, the university required students to complete 75 h (100 class hours) of volunteer service and document it using its new Second Classroom reporting system, and their hours of volunteer service should have been calculated according to the records on Volunteer Beijing [41]. This made the system indispensable for students at this university and further made the university a typical case for research on this topic, as it is the university in Beijing with the longest history of the integrated use of Volunteer Beijing, and has the richest and most complete records of volunteer service.

The study uses three sets of data, divided into three main sections. The paper discusses the sustainable value of the volunteer service of contemporary Chinese university students using three dimensions: measurable ability, evaluable inner feeling, and evaluable external responsibility. The first set of data used is a special sample set of experimental and control groups with basically consistent ethnographic characteristics over the same period. The second set of data includes questionnaire data obtained by strict equidistant stratified sampling. The third set of data includes the average annual volunteer time of students and their self-rating of responsibility from the above-mentioned questionnaire data. Through the microanalysis of the three sets of "black box" data, the paper studies the sustainable impact of volunteer service on university students, including their academic performance, employment results, ability acquisition results, and improvements in their sense of responsibility.

### 3.1. Analysis of the Impact of Volunteer Service Time on Academic Performance and Employment

3.1.1. Selection of Independent Variables

In this part, students' volunteer service time as recorded by the information system was selected as the independent variable. Starting in 2012, undergraduates at the sample university have been asked to participate in a variety of volunteer services. Their volunteer service hours have been accumulated year by year until their senior year. Besides participating in volunteer service every year, students at Faculty A are required to complete more hours of volunteer service than other faculties, starting with the 2015 class. Social contribution points (see Table 1) have been awarded with reference to the Revised Rules of Comprehensive Student Evaluation of 2015, and are based on levels of service time, from 30 h to 120 h or more. This rule has been implemented since the 2015 class and does not apply to students of the 2012, 2013, or 2014 classes.

**Table 1.** Revised rules for points to be awarded for volunteer service time in Faculty A.

| Volunteer Service Time | Points |
| --- | --- |
| 30–44 h | 1 point |
| 45–59 h | 1.5 points |
| 60–74 h | 2 points |
| 75–89 h | 2.5 points |
| 90–104 h | 3 points |
| 105–119 h | 3.5 points |
| 120 h and above | 4 points |

### 3.1.2. Selection of Dependent Variables

The study focuses on the impacts of volunteer service participation on students' academic achievement and employment as dependent variables.

Studies by Robert C. Serow and Alexander W. Astin show that students' participation in volunteer service has a positive impact on their academic performance during their studies [11,23]. Therefore, this paper first focuses on the average *GPA* of students in each academic year, which was calculated using all the subject scores of students from the 2012–2015 classes according to the following method:

$$Average\ GPA = \frac{GPA\ per\ academic\ year}{\sum Number\ of\ credits} = \frac{Credit \times GPA\ coefficient}{Number\ of\ credits} \quad (1)$$

Studies by Robert C. Serow, Alexander W. Astin, and Linda J. Sax also show that volunteer service participation has a significant impact on students' leadership development and career choices [7,23]. Therefore, the second dependent variable selected in this study was employment. According to data about the employment types of graduates and interviews with employment authorities, the types of employment of Chinese university graduates, ranked from high quality to low quality, have been as follows: employment agreements (including enlistment in the army and the Far West Program), labor contracts, self-employment, employment certificates, freelancing, and waiting to be employed. Since "going to graduate school at home" and "going to graduate school abroad" are both options for further study, rather than employment, the relevant data are not represented in the analysis.

The scholarship policy incentive of a comprehensive evaluation, as was briefly mentioned above, was defined as a reform interaction item. The analysis of such evaluations' impacts on total volunteer service time and students' academic achievement and employment will help determine the real effect of the reform policy. Finally, necessary control variables were added to ensure the validity of the data analysis.

### 3.1.3. The Analysis Method

This section uses the data on volunteer service time of four classes of undergraduates, from 2012 to 2015, at the sample university, the average *GPA* of each student per academic year as calculated with the raw data of their First Classroom *GPA* according to the *GPA* calculation formula, and the employment data of the 2015 class students. Since University A had adopted the new incentive policy for its 2015 class students, whereas the 2012–2014 classes' students are not affected, a natural experiment could be conducted with the two groups. Therefore, the authors constructed a difference-in-differences model for empirical analysis. The demonstration equation is as follows:

$$Y_{it} = \beta_0 + \beta_1 D_t + \beta_2 D_{treat} + \beta_3 D_t \cdot D_{treat} + \alpha X_{it} \ (i = 1, \cdots, n; t = 1, 2) \quad (2)$$

In the equation, *i* represents the students and *t* represents the year. $D_{treat}$ represents whether it is a dummy variable of university students affected by the policy change, with 1 indicating "yes" and 0 indicating "no". $D_t$ represents the time dummy variable, with 1 meaning after students are affected by the policy and 0 meaning before students are

affected. $X_{it}$ represents a series of control variables, including year of graduation, gender, ethnicity, major, and university, which are all added to the final regression model. The core estimation coefficient in this paper is $\beta_3$, which reflects the impact of the new incentive policy on student achievement. In order to overcome the influence of individual differences and policies related to different time nodes on the estimation results, this research further controlled for the fixed effects of individuals and time. The final demonstration equation is as follows:

$$Y_{it} = \beta_0 + \beta_1 D_t + \beta_2 D_{treat} + \beta_3 D_t \cdot D_{treat} + \alpha X_{it} + \delta_i + \gamma_t + \varepsilon_{it} (i = 1, \cdots, n; t = 1, 2) \quad (3)$$

*3.2. Comparative Analysis of University Students' Expectations and Experiences of Their Ability Acquisition in Volunteer Service*

From the existing literature, we can see that students participate in voluntary service through material incentives, classroom goals, and comprehensive evaluation incentives [25–27,42]. Their participation in volunteer service also helps to improve their leadership and social responsibility [7,11]. To further discuss the effectiveness of volunteer service on students, the study used the results of a questionnaire conducted in early 2018 on the self-description of the abilities of students at the sample university to discuss the growth of their actual abilities and any increase in their sense of social responsibility through participation in volunteer service. In the questionnaire, the scale is designed to measure students' ability acquisition, team cooperation, leadership, etc.

First, the abilities related to volunteer service were categorized into three dimensions, according to the previous research: personal ability, teamwork, and leadership management. The three dimensions further fall into 18 items, measured in the form of a 5-point Likert scale, i.e., "completely inconsistent", "inconsistent", "neither consistent nor in-consistent", "consistent" and "completely consistent", counting for 1, 2, 3, 4, and 5 point(s), respectively.

The personal ability dimension refers to the overall framework and essence of the Key Competences for the Development of Chinese Students published by the Key Competence Research Group in 2016, i.e., cultural foundation, self-development and social participation, and further extends to seven aspects: humanistic literacy, scientific thinking, self-management, emotional regulation, language proficiency, learning ability, and practical ability [43] (See Table 2).

**Table 2.** Items in the personal ability scale.

| Measurement Dimensions | Items |
| --- | --- |
| Personal ability | (1) Humanistic literacy: I hope to learn certain cultural knowledge and improve humanistic literacy through volunteer activities. (2) Scientific thinking: I hope to exercise logical thinking through voluntary activities and analyze problems from multiple perspectives and dialectically. (3) Self-management: I hope to understand myself better and manage my time and energy more reasonably through voluntary activities. (4) Emotional regulation: I hope to learn to adjust myself physically and mentally through voluntary activities. (5) Language proficiency: I hope to improve my ability to express myself in Chinese and English or command other minor languages through voluntary activities. (6) Learning ability: I hope to learn to gain new knowledge and methods more actively through voluntary activities and draw lessons from the learning process. (7) Practical ability: I hope to enhance my execution through volunteer activities, and sometimes to enhance my understanding of my area of expertise and apply such knowledge to practice. |

Teamwork ability draws on concepts from the study of organization. In the context of the actual demands on university students in the current era, the teamwork ability that university students hope to improve by participating in voluntary activities can be divided into six aspects: self-awareness, perspective-taking, rationality, consciousness of principle, collective consciousness, and overall consciousness (See Table 3).

**Table 3.** Items in teamwork motivation scale.

| Measurement Dimensions | Items |
|---|---|
| Teamwork ability | (1) Self-awareness: I hope to find my own strengths and weaknesses by participating in various types of volunteer activities so that I can put my strengths to better use and know where I fit in the team and achieve better development. <br> (2) Perspective-taking: I hope to learn perspective-taking by dealing with different people in volunteer activities so that I can be more sensitive to others' feelings and needs, become more understanding and supportive, and contribute more effectively. <br> (3) Rationality: Through volunteer activities, I hope to learn to deal with disputes with others more rationally and calmly, and communicate instead of quarreling with them, so as to improve teamwork efficiency. <br> (4) Consciousness of principle: I hope to stand my ground and stick to my own opinions and positions when handling complicated situations in volunteer activities. <br> (5) Collective consciousness: I hope to better integrate into my team, consciously observe the rules of my team, and improve my sense of responsibility, honor, and pride towards my team by participating in volunteer activities. <br> (6) Overall consciousness: I hope to think in terms of the big picture and make decisions that are forward-looking and in line with the greater good. |

The dimensions of leadership management draw on the five-force model of leadership proposed by the research group of the Chinese Academy of Sciences in 2006. This study cites the "five forces" (charisma, foresight, appeal, decisiveness, and control) as the indicators of leadership [44] (See Table 4).

**Table 4.** Items in leadership scale.

| Measurement dimensions | Items |
|---|---|
| Leadership | (1) Charisma: I hope to become more confident, passionate, and charismatic through volunteer activities. <br> (2) Foresight: I hope to improve my foresight through volunteer activities and learn to plan ahead and manage my tasks better. <br> (3) Appeal: I hope to learn to exert a positive appeal on others and set a good example for them. <br> (4) Decisiveness: I hope to become more decisive, independent, and assertive through voluntary activities. <br> (5) Control: I hope to learn to take control through volunteer activities. |

This section mainly describes the use of factor extraction and variance analysis methods. Factor analysis was mainly used to analyze the ability acquisition expectations and acquisition experiences of volunteers engaged in voluntary service, and further explored the differences between them. Variance analysis reconsidered the influence of various attributes of the research object on students' ability acquisition expectations and acquisition experiences.

*3.3. Analysis of the Influence of Volunteer Service on University Students' Sense of Responsibility*

3.3.1. Scale Design of Sense of Responsibility

(1)　Personal responsibility

Based on the existing research, the study included eight items in the scale of personal responsibility, including "self-survival", "self-positioning", "self-planning", "self-management", "self-reflection", "self-improvement" and "self-innovation", according to the characteristics of contemporary university students and the meaning of personal responsibility (Table 5).

**Table 5.** Items in personal responsibility scale.

| Measurement Dimensions | Items |
| --- | --- |
| Personal responsibility | (1) Self-survival: I love life and value health and safety. (2) Self-positioning: I have a deep understanding and clear positioning of myself. (3) Self-planning: I have a clear plan for my life and practice it seriously. (4) Self-management: I have good living and study habits, and being punctual and efficient is my strength. (5) Self-reflection: I always practice regular self-reflection and introspection. (6) Self-improvement: I always actively improve my abilities and expand my horizons, in the hope of continuously improving myself. (7) Self-innovation: I am always willing to think outside the box and explore the unknown. (8) Self-realization: I make unremitting efforts for my life and approach my goal step by step. |

Based on the idea of "being responsible for yourself", the personal responsibility scale follows the logic of self-awareness, self-development, and self-improvement.

In terms of self-awareness, "self-survival" is the prerequisite for understanding oneself. Loving oneself and being responsible for one's own life is the foundation of living and the origin of personal responsibility. On this basis, "self-positioning" stands for a further understanding of oneself, and only by finding one's own position can one achieve self-development and improvement.

In terms of self-development, planning, management, and reflection are necessary steps for developing oneself. "Self-planning" means being responsible for one's own life, making a good plan, and putting it into practice; "Self-management" refers to students' abilities to effectively discipline themselves and manage their own lives and studies; "Self-reflection" refers to the ability to think independently, reflect on oneself, and make continuous improvement.

In terms of self-improvement, university students need to pursue higher goals. To actively enrich one's knowledge and ability is a reflection of "self-improvement". To be pioneering and innovative and daring to make breakthroughs embodies the courage to challenge oneself, a reflection of "self-innovation". To have firm ideals and beliefs and pursue demands at the highest level—"self-realization"—is the ultimate goal of self-improvement.

(2)　Collective responsibility

The collective responsibility scale mainly refers to the socialist core values combined with the realities of contemporary university students, including eight items: "building a prosperous and strong country", "building a democratic country", "ensuring civilization and harmony", "ensuring freedom and equality", "protecting justice and the law", "loving the motherland"; "being dedicated and eager to learn"; and "being honest and friendly" (Table 6).

**Table 6.** Items in collective responsibility scale.

| Measurement Dimensions | Items |
|---|---|
| Collective responsibility | (1) Building a prosperous and strong country: I have lofty goals in life and hope to contribute to the prosperity and strength of my motherland. <br> (2) Building a democratic country: I have good political awareness and pursue socialist democracy, where the people are the masters of the country. <br> (3) Ensuring civilization and harmony: I try my best to adjust social conflicts and promote social civilization and harmony. <br> (4) Ensuring freedom and equality: I respect the freedom of others, pursue equality in society, and seek common ground while reserving differences. <br> (5) Protecting justice and law: I pursue and support fairness and justice, strictly abide by laws and regulations, use legal knowledge to safeguard rights and interests, and never cover up others' violations of law and discipline. <br> (6) Loving the motherland: I have a strong sense of national pride, regard the revitalization of the Chinese nation as my duty, actively promote national unity, and safeguard national sovereignty and the reunification of my motherland. <br> (7) Being dedicated and eager to learn: As a student, I take my studies seriously and love learning about science and culture, so that I can better give back to society in the future. <br> (8) Being honest and friendly: I value honesty, keep my promises, refrain from harming the interests of others, and treat people in a friendly manner. |

Socialist core values are divided into three levels: national, social, and personal. "Prosperity", "democracy", "civilization" and "harmony" are the pursuits of socialist core values at the national level [45]. For contemporary university students, they are embodied by three measurement indicators: "building a prosperous and strong country", "building a democratic country", and "ensuring civilization and harmony". "Building a prosperous and strong country" refers to university students' personal beliefs in developing themselves for their country's prosperity and strength; "building a democratic country" measures the ability to be the backbones of their own country and enhance political awareness; "ensuring civilization and harmony" measures the sense of responsibility to attend to their surroundings and promote national civilization and harmony.

"Freedom", "equality", "justice" and "rule of law" are the pursuits of socialist core values at the social level. For university students, they are embodied as "ensuring freedom and equality" and "protecting justice and the law" [45]. "Ensuring freedom and equality" measures whether university students respect social diversity; "protecting justice and law" highlights that university students should protect justice, respect the law, support the rule of law society, and be law-abiding citizens.

"Patriotism", "dedication", "honesty" and "friendliness" are the norms of socialist core values at the personal level [45]. For contemporary university students, they are mainly divided into "loving the motherland", "being dedicated and eager to learn", and "being honest and friendly". "Loving the motherland" is the patriotic spirit that must be pursued by all university students and Chinese people; "being dedicated and eager to learn" reflects that contemporary university students should be dedicated to their work and focus on their studies; "being honest and friendly" emphasizes that university students should be honest and trustworthy, and be kind to others.

3.3.2. The Analysis Method

(1) Regression analysis of the average annual volunteer service time and the total self-rating of responsibility was performed using the following formula:

$$Res = \beta_0 + \alpha Adur + \beta_1 x_1 + \beta_2 x_2 + \ldots \beta_p x_p + \varepsilon \tag{4}$$

in which *Res* represents the comprehensive sense of responsibility score, *Adur* represents the annual average volunteer time, and $x_1$ to $x_p$ represent a series of control variables, including gender, family province, grade, faculty, whether a student was an only child, whether a student was a former student leader, family economic status, and political status. Control variables were added one by one.

(2)     Regression analysis of average annual volunteer service time and self-rating of personal responsibility was performed using the following formula:

$$Ego = \beta_0 + \alpha Adur + \beta_1 x_1 + \beta_2 x_2 + \ldots \beta_p x_p + \varepsilon \tag{5}$$

in which *Ego* represents the comprehensive sense of responsibility score, *Adur* represents the annual average volunteer time, and $x_1$ to $x_p$ represent a series of control variables, including gender, family province, grade, facultyfaculty, whether a student was an only child, whether a student was a former student leader, family economic status, and political status. Control variables were added one by one.

(3)     Regression analysis of the average annual volunteer service time and self-rating of collective responsibility was performed using the following formula:

$$Col = \beta_0 + \alpha Adur + \beta_1 x_1 + \beta_2 x_2 + \ldots \beta_p x_p + \varepsilon \tag{6}$$

in which *Col* represents the comprehensive sense of responsibility score, *Adur* represents the annual average volunteer time, and $x_1$ to $x_p$ represent a series of control variables, including gender, family province, grade, faculty, whether a student was an only child, whether a student was a former student leader, family economic status, and political status. Control variables were added one by one.

## 4. Results

### *4.1. Impact of Volunteer Service on University Students' Academic Performance and Employment*

#### 4.1.1. Descriptive Statistics of Variables

In the samples of the 2012–2015 classes, since the students were asked to complete 75 h of volunteer service before graduation, the data recorded in the system for less than 75 h, generally due to untimely or missing records of volunteer organizers, was excluded. For students whose volunteer time was for a longer period of time than the scale measured, the effectiveness of volunteer service projects was manually screened, and the samples whose service times were obviously abnormal were deleted. In addition, since the maximum value of the average *GPA* is 4, the abnormal samples whose average *GPA* exceeded 4 were deleted. Furthermore, samples whose First-Class records were empty were also excluded. Finally, data on 28,606 samples for the four classes were obtained, as shown in Table 2. Among them, "total time" refers to the total number of hours that students spent on volunteer service in four years, with the minimum value being 75 h (based on the minimum graduation requirements of the sample university), the maximum value being 839 h, and the average being 120.815 h. The *GPA* per year refers to the average *GPA* of each student in the four years, with the minimum value being 0.522, the maximum value being 3.987, and the average being 3.333. Employment was assigned values according to different types of employment, i.e., employment agreements (including enlistment in the army and the Far West Program), labor contracts, self-employment, employment certificates, freelancing, and waiting to be employed, with 6 assigned to employment agreements and 1 assigned to waiting for employment, and the average being 1.6008. The Faculty A reform interaction was defined as the new incentive policy of Faculty A, with 1 being affected by the reform and 0 being not affected (Table 7).

#### 4.1.2. Impact of Faculty A's Reform Policy on the Total Volunteer Service Time

Faculty A implemented a volunteer time incentive reform for the 2015 freshman class in 2015. It should be noted that there is a major difference between this rule at Faculty A and the national service time record and certification rule for all university students: the volunteer service time record and certification rule is merely a mandatory measure for

meeting Second Classroom requirements and is not linked to any award or evaluation. The award point policy at Faculty A, however, acts as an incentive for awards and evaluation: the more points students get in the comprehensive evaluation, the higher grade they will receive. Therefore, this paper takes all the students from the 2012 to 2015 classes as participants, with the graduates who were subject to the policy as the experimental group, and all the other students as the control group. The difference-in-differences method was used to determine the changes in students' academic achievements due to the impact of the policy. Whether or not the students of the sample university were affected by the policy, they all needed to complete 75 h of volunteer service according to the Second Classroom requirements within four years before graduation. Therefore, the data analysis described in this section used the excess of total volunteer service time beyond 75 h.

**Table 7.** Descriptive statistics.

| Variables | Number of Observations | Mean Value | Standard Deviation | Minimum Value | Maximum Value |
|---|---|---|---|---|---|
| Total time | 28,606 | 120.815 | 72.273 | 75 | 839 |
| *GPA* | 28,606 | 3.333 | 0.399 | 0.522 | 3.987 |
| Employment | 28,606 | 1.601 | 2.238 | 0 | 6 |
| Faculty A reform interaction item | 28,606 | 0.032 | 0.177 | 0 | 1 |

The authors first took a look at the impact of Faculty A's reform on the total volunteer service time. The following table presents the regression models of Faculty A's reform on the total volunteer service time. Model 1 controlled for the year of graduation, model 2 controlled for the year of graduation and gender, model 3 controlled for the year of graduation, gender, and ethnicity, model 4 controlled for the year of graduation, gender, ethnicity, and major, and model 5 controlled for the year of graduation, gender, ethnicity, major and faculties. Since there were too many control variables for majors and colleges, which affects the presentation of data, the following table only shows the regression of constant item $\beta_0$ and the interaction coefficient $\beta_3$. The following conclusions can be drawn from the regression results(Table 8):

**Table 8.** Regression results of Faculty A's reform policy on total volunteer service time.

|  | (1) | (2) | (3) | (4) | (5) |
|---|---|---|---|---|---|
|  | Total time | Total time | Total time | Total time | Total time |
| Faculty A reform interaction | 58.283 *** | 58.210 *** | 58.186 *** | 70.364 *** | 70.613 *** |
|  | (4.114) | (4.108) | (4.082) | (4.245) | (4.249) |
| Constant term | 122.141 *** | 120.823 *** | 122.062 *** | 123.914 *** | 111.949 *** |
|  | (0.900) | (1.173) | (5.086) | (7.473) | (8.202) |
| Year of graduation | Control | Control | Control | Control | Control |
| Gender |  | Control | Control | Control | Control |
| Ethnicity |  |  | Control | Control | Control |
| Major |  |  |  | Control | Control |
| Faculty |  |  |  |  | Control |
| N | 28,606 | 28,606 | 28,606 | 28,606 | 28,606 |
| R-sq | 0.037 | 0.037 | 0.042 | 0.063 | 0.067 |
| adj. R-sq | 0.036 | 0.036 | 0.039 | 0.058 | 0.062 |

Note: * $p < 0.1$, ** $p < 0.05$, *** $p < 0.01$. The more asterisks, the higher the level of significance.

Firstly, the goodness of fit of the model gradually improved with the addition of control variables, particularly after the addition of the control variables major and faculty.

Secondly, the coefficient of the reform interaction is relatively stable and very significant. After the control variables major and faculty were added, the coefficient became larger. $\beta_3$ in model 5 is 70.613, which means that students affected by the policy spent 70.613 h more on volunteer service than students not affected.

### 4.1.3. Impact of Faculty A's Reform Policy on the Average GPA of Students per Year

The authors then analyzed the impact of Faculty A's reform policy on the average *GPA* of students per year. Similarly, model 1 controlled for the year of graduation, model 2 controlled for the year of graduation and gender, model 3 controlled for the year of graduation, gender, and ethnicity, model 4 controlled for the year of graduation, gender, ethnicity, and major, and model 5 controlled for the year of graduation, gender, ethnicity, major, and faculty. The table mainly presents the regression of constant item $\beta_0$ and the interaction coefficient $\beta_3$. The following conclusions can be drawn from the regression results (Table 9):

**Table 9.** Regression results of Faculty A's reform policy on students' average *GPA* per year.

|  | (1) | (2) | (3) | (4) | (5) |
|---|---|---|---|---|---|
|  | *GPA* | *GPA* | *GPA* | *GPA* | *GPA* |
| Faculty A's reform interaction | 0.100 *** | 0.094 *** | 0.093 *** | 0.030 ** | 0.033 ** |
|  | (0.0112) | (0.012) | (0.012) | (0.015) | (0.014) |
| Constant term | 3.423 *** | 3.318 *** | 3.410 *** | 3.265 *** | 3.509 *** |
|  | (0.005) | (0.006) | (0.040) | (0.048) | (0.060) |
| Year of graduation | Control | Control | Control | Control | Control |
| Gender |  | Control | Control | Control | Control |
| Ethnicity |  |  | Control | Control | Control |
| Major |  |  |  | Control | Control |
| Faculty |  |  |  |  | Control |
| N | 28,606 | 28,606 | 28,606 | 28,606 | 28,606 |
| R-sq | 0.053 | 0.080 | 0.096 | 0.130 | 0.145 |
| adj. R-sq | 0.052 | 0.079 | 0.093 | 0.125 | 0.140 |

Note: * $p < 0.1$, ** $p < 0.05$, *** $p < 0.01$. The more asterisks, the higher the level of significance.

Firstly, the goodness of fit of the model gradually improved with the addition of control variables, particularly after the addition of the control variables major and faculty.

Secondly, the coefficient of the reform interaction is relatively stable and significant. After the control variables major and faculty were added, the coefficient became slightly smaller and less significant. However, in general, the students affected by the policy had a higher average *GPA* compared with those not affected, with an increase of 0.033, which represents better academic performance.

### 4.1.4. Impact of Faculty A's Reform Policy on Employment

Similarly, model 1 controlled for the year of graduation, model 2 controlled for the year of graduation and gender, model 3 controlled for the year of graduation, gender, and ethnicity, model 4 controlled for the year of graduation, gender, ethnicity, and major, and model 5 controlled for the year of graduation, gender, ethnicity, major, and faculty. The table mainly presents the regression of constant item $\beta_0$ and the interaction coefficient $\beta_3$. Since students can either pursue further study or find jobs after graduation, and since further study is not strictly considered to be employment, students who pursued further study were excluded, and we observed the changes in the employment of the 10,828 remaining samples affected by the policy. The following conclusions can be drawn from the regression results (Table 10):

Firstly, the goodness of fit of the model gradually improved with the addition of control variables, particularly after the addition of the control variables major and university.

Secondly, the coefficient of the reform interaction is relatively stable and significant. When the control variables major and university were included, the coefficient decreased. However, in general, the longer the volunteer service time, the better the employment, and the average employment score of students affected by the policy increased by 0.343.

**Table 10.** Regression results of Faculty A's reform policy on employment.

|  | (1) | (2) | (3) | (4) | (5) |
|---|---|---|---|---|---|
|  | Employment | Employment | Employment | Employment | Employment |
| Faculty A's reform interaction | 0.695 *** | 0.697 *** | 0.706 *** | 0.349 *** | 0.343 *** |
|  | (0.083) | (0.083) | (0.083) | (0.092) | (0.092) |
| Constant term | 4.367 *** | 4.491 *** | 4.787 *** | 5.220 *** | 5.225 *** |
|  | (0.028) | (0.034) | (0.245) | (0.277) | (0.276) |
| Year of graduation | Control | Control | Control | Control | Control |
| Gender |  | Control | Control | Control | Control |
| Ethnicity |  |  | Control | Control | Control |
| Major |  |  |  | Control | Control |
| Faculty |  |  |  |  | Control |
| N | 10,828 | 10,828 | 10,828 | 10,828 | 10,828 |
| R-sq | 0.044 | 0.047 | 0.063 | 0.148 | 0.149 |
| adj. R-sq | 0.041 | 0.044 | 0.057 | 0.137 | 0.138 |

Note: * $p < 0.1$, ** $p < 0.05$, *** $p < 0.01$. The more asterisks, the higher the level of significance.

### 4.2. Comparative Analysis of University Students' Expectations and the Results of Ability Acquisition in Volunteer Service

#### 4.2.1. Descriptive Statistics

The survey stratified students according to their student IDs using a strict stratified sampling method. A total of 854 questionnaires were distributed to all the students in the 2013–2017 classes in every faculty of the sample university. A total of 806 questionnaires were collected, of which 29 were invalid and 777 were valid. Through participation, the questionnaire primarily included descriptions of the basic demographic characteristics of students, as well as the ability growth scale and responsibility growth scale. The major descriptive statistics, apart from demographic characteristics, are shown in Table 11. Among them, the minimum *GPA* is 1.370, and the maximum *GPA* is 3.960, with the average being 3.215. The average annual volunteer service time is 35.613 h. Personal ability, teamwork ability, and leadership ability self-rating values are 3.869, 4.005, and 3.798, respectively; personal responsibility, collective responsibility, and overall responsibility scores are 4.002, 4.167, and 4.085, respectively.

**Table 11.** Descriptive statistics of questionnaire data.

| Variables | Observation Value | Mean Value | Standard Deviation | Minimum Value | Maximum Value |
|---|---|---|---|---|---|
| *GPA* | 777 | 3.215 | 0.475 | 1.370 | 3.960 |
| Average volunteer service time per year | 777 | 35.613 | 26.590 | 0 | 186 |
| Personal ability score | 777 | 3.869 | 0.572 | 1 | 5 |
| Teamwork ability score | 777 | 4.005 | 0.544 | 1 | 5 |
| Leadership score | 777 | 3.798 | 0.633 | 1 | 5 |
| Personal responsibility score | 777 | 4.002 | 0.706 | 1 | 5 |
| Collective responsibility score | 777 | 4.167 | 0.695 | 1 | 5 |
| Overall responsibility score | 777 | 4.085 | 0.667 | 1 | 5 |

#### 4.2.2. Comparative Analysis

By comparing the three comprehensive dimensions of volunteer ability acquisition expectations and results, the study revealed that the supply of personal ability and leadership was less than the demand, while the supply of teamwork ability was greater than the demand (See Table 12).

**Table 12.** The supply and demand of abilities of university student volunteers.

| Category | Number of Students who Want to Improve This Ability the Most | Proportion | Number of Students Who Improved This Ability the Most | Proportion |
|---|---|---|---|---|
| Personal ability | 423 | 54.44% | 396 | 50.97% |
| Teamwork ability | 253 | 32.56% | 297 | 38.22% |
| Leadership | 101 | 13.00% | 84 | 10.81% |

The authors conducted an exploratory factor analysis of all the dimensions involved when measuring the 18 micro-dimensions in order to further refine the ability acquisition expectations and experiences. Ability acquisition expectations were analyzed, with results shown in Table 13, and the results of the ability acquisition analysis are shown in Table 14. From the perspective of factor extraction, all the projects involving personal motivation and teamwork were classified into the same category, and the projects involving leadership management were classified into another category. The former category mainly involves the individual's ability to complete tasks, while the latter mainly involves the ability to lead teams to complete tasks.

**Table 13.** The factor load matrix after rotation for ability acquisition expectations.

| Personal Motivation | Factor Analysis | | Teamwork Motivation | Factor Analysis | | Leadership and Management Motivation | Factor Analysis | |
|---|---|---|---|---|---|---|---|---|
| | 1 | 2 | | 1 | 2 | | 1 | 2 |
| Humanistic accomplishment | **0.756** | 0.287 | Self-awareness | **0.726** | 0.460 | Charisma | 0.321 | **0.816** |
| Scientific thinking | **0.699** | 0.350 | Perspective taking | **0.685** | 0.455 | Foresight | 0.412 | **0.779** |
| Self-management | **0.798** | 0.266 | Rationality | **0.643** | 0.530 | Appeal | 0.362 | **0.737** |
| Emotional regulation | **0.756** | 0.261 | Consciousness of principle | **0.624** | 0.534 | Decisiveness | 0.380 | **0.802** |
| Language proficiency | **0.552** | 0.467 | Collective consciousness | **0.693** | 0.471 | Control | 0.338 | **0.793** |
| Learning ability | **0.727** | 0.415 | Overall consciousness | **0.653** | 0.539 | | | |
| Practical ability | **0.712** | 0.398 | | | | | | |

Note: The matrix is orthogonally rotated, the KMO value is 0.964, and the significance of the Bartlett sphericity test is less than 0.001. The reliability of each subscale was analyzed using the SPSS20.0 Cronbach α value, and the data presented are all greater than 0.7. The Cronbach α value of the total scale is as high as 0.966, indicating that the reliability of the data is high. The bold factor scores are higher than the other factor in the same motivation, indicating that common factor 1 is more related to individual motivation and team cooperation motivation, and common factor 2 is more related to leadership and management motivation.

**Table 14.** The factor load matrix after rotation for ability acquisition results.

| Personal Ability | Factor Analysis | | Teamwork Ability | Factor Analysis | | Leadership and Management Ability | Factor Analysis | |
|---|---|---|---|---|---|---|---|---|
| | 1 | 2 | | 1 | 2 | | 1 | 2 |
| Humanistic accomplishment | **0.684** | 0.393 | Self-awareness | **0.776** | 0.410 | Charisma | 0.397 | **0.804** |
| Scientific thinking | **0.615** | 0.509 | Rationality | **0.777** | 0.359 | Foresight | 0.438 | **0.767** |
| Self-management | **0.779** | 0.368 | Perspective taking | **0.758** | 0.435 | Appeal | 0.417 | **0.781** |
| Emotional regulation | **0.732** | 0.365 | Consciousness of principle | **0.731** | 0.462 | Decisiveness | 0.465 | **0.758** |
| Language proficiency | **0.522** | 0.555 | Collective consciousness | **0.795** | 0.372 | Control | 0.381 | **0.830** |
| Learning ability | **0.669** | 0.503 | Overall consciousness | **0.745** | 0.423 | | | |
| Practical ability | **0.670** | 0.469 | | | | | | |

Note: The matrix is orthogonally rotated, the KMO value is 0.964, and the significance of the Bartlett sphericity test is less than 0.001. The reliability of each subscale was analyzed by using the SPSS20.0 Cronbach α value, and the data presented are all greater than 0.7. The Cronbach α value of the total scale is as high as 0.972, indicating that the reliability of the data is high. The bold factor scores are higher than the other factor in the same motivation, indicating that common factor 1 is more related to individual motivation and team cooperation motivation, and common factor 2 is more related to leadership and management motivation.

After factor extraction, the authors conducted an ANOVA on students' personal attributes and ability acquisition expectations and experiences. The findings are shown in Table 15. The "faculty" attribute had more impact on students' ability acquisition expectations and experiences, while the only child, political status, and student leader experiences had more impact on students' ability acquisition results.

**Table 15.** The relationship between students' personal attributes and ability acquisition expectations and experiences.

| Factor | Ability Acquisition Expectations | | | Ability Acquisition Results | | |
|---|---|---|---|---|---|---|
| | Personal and Collective Ability | Leadership and Management Ability | Comprehensive Ability | Personal and Collective Ability | Leadership and Management Ability | Comprehensive Ability |
| Gender | 0.080 | 0.844 | 0.273 | 0.923 | 0.512 | 0.757 |
| Faculty | **0.044** | 0.097 | **0.048** | **0.024** | 0.333 | 0.107 |
| Grade | 0.083 | 0.184 | 0.100 | 0.095 | 0.404 | 0.200 |
| Only child or not | 0.314 | 0.722 | 0.749 | **0.018** | **0.006** | **0.007** |
| Family socioeconomic position | 0.535 | 0.311 | 0.386 | 0.507 | 0.058 | 0.172 |
| Family province | 0.595 | 0.996 | 0.952 | 0.715 | 0.917 | 0.867 |
| Political status | 0.150 | 0.192 | 0.154 | **0.027** | **0.012** | **0.013** |
| Student leader experiences | 0.454 | 0.662 | 0.607 | 0.102 | **0.044** | 0.057 |

Note: The raw score of comprehensive ability is represented as the average score of personal ability, teamwork ability, and leadership ability. Bold numbers indicate $p < 0.05$.

Knight and Yorke, an employability research authority in the UK, put forward the USEM Model [46], consisting of (1) the Understanding of the area of expertise; (2) professional and general Skills required for work; (3) Efficacy Beliefs, which include personal qualities, self-confidence and enjoyment of learning; (4) and Meta-cognition, which reflects strategic response and thinking [47]. As students' ability acquisition in this study generally overlaps with this model, the improvement in students' abilities through volunteer service is also an improvement in their employability. The above studies show that students have a strong potential desire to improve their employability by participating in voluntary service, but there is still a gap in their actual experience. Although this study proves that more participation in voluntary service under relevant policy incentives has a positive impact on employment, such improvement may be negatively affected by students' actual experiences. Further empirical analysis is required in the future.

### 4.3. Impact of Volunteer Service on University Students' Sense of Responsibility

4.3.1. Regression Analysis of Average Annual Volunteer Service Time and Total Self-Rating of Sense of Responsibility

The regression results show that the goodness of fit of the model gradually increased, and the regression coefficient of the average annual time to a sense of social responsibility gradually became more significant. The highest coefficient was achieved when only the gender, province, grade, faculty, and only-child status were controlled for. The most significant coefficient was achieved when the student leader experience was added as a control variable on the basis of the above five items (Table 16).

4.3.2. Regression Analysis of the Average Annual Volunteer Service Time and the Self-Rating of Personal Responsibility

The regression results show that the goodness of fit of the model gradually increased, and the regression coefficient of the annual average time to the sense of personal responsi-

bility gradually became more significant. The highest coefficient was also achieved when gender, province, grade, faculty, and only-child status were controlled for, and the most significant coefficients were achieved in the following two scenarios: one was when gender, province, grade, and faculty were controlled for, and the other was when gender, province, grade, faculty, only-child status, and student leader experience were controlled for (Table 17).

**Table 16.** Regression results of average time to total self-rating of responsibility.

|  | (1) | (2) | (3) | (4) | (5) | (6) | (7) | (8) | (9) |
|---|---|---|---|---|---|---|---|---|---|
| Average annual time | 0.001 | 0.001 | 0.001 | 0.001 | 0.002 * | 0.003 * | 0.002 *** | 0.002 ** | 0.002 ** |
|  | (0.001) | (0.001) | (0.001) | (0.001) | (0.001) | (0.001) | (0.001) | (0.001) | (0.001) |
| Constant term | 4.035 *** | 4.020 *** | 3.993 *** | 4.109 *** | 3.996 *** | 4.150 *** | 4.258 *** | 4.443 *** | 4.544 *** |
|  | (0.040) | (0.099) | (0.179) | (0.185) | (0.224) | (0.254) | (0.266) | (0.326) | (0.475) |
| Gender |  | Control | Control | Control | Control | Control | Control | Control | Control |
| Province |  |  | Control | Control | Control | Control | Control | Control | Control |
| Grade |  |  |  | Control | Control | Control | Control | Control | Control |
| Faculty |  |  |  |  | Control | Control | Control | Control | Control |
| Only child or not |  |  |  |  |  | Control | Control | Control | Control |
| Student leader experiences |  |  |  |  |  |  | Control | Control | Control |
| Family socioeconomic position |  |  |  |  |  |  |  | Control | Control |
| Political status |  |  |  |  |  |  |  |  | Control |
| N | 777 | 777 | 777 | 777 | 777 | 777 | 777 | 777 | 777 |
| $R^2$ | 0.003 | 0.003 | 0.042 | 0.049 | 0.089 | 0.095 | 0.097 | 0.105 | 0.108 |
| Adjusted $R^2$ | 0.002 | 0.001 | −0.002 | 0.002 | 0.029 | 0.034 | 0.035 | 0.039 | 0.038 |

Note: * $p < 0.1$, ** $p < 0.05$, *** $p < 0.01$.

**Table 17.** Regression results of average annual time to personal responsibility.

|  | (1) | (2) | (3) | (4) | (5) | (6) | (7) | (8) | (9) |
|---|---|---|---|---|---|---|---|---|---|
| Average annual time | 0.001 | 0.001 | 0.001 | 0.002 | 0.002 *** | 0.003 * | 0.002 *** | 0.002 ** | 0.002 ** |
|  | (0.001) | (0.001) | (0.001) | (0.001) | (0.001) | (0.001) | (0.001) | (0.001) | (0.001) |
| Constant term | 3.958 *** | 3.941 *** | 3.928 *** | 4.066 *** | 4.009 *** | 4.219 *** | 4.372 *** | 4.660 *** | 4.797 *** |
|  | (0.042) | (0.104) | (0.189) | (0.196) | (0.260) | (0.269) | (0.282) | (0.346) | (0.503) |
| Gender |  | Control | Control | Control | Control | Control | Control | Control | Control |
| Province |  |  | Control | Control | Control | Control | Control | Control | Control |
| Grade |  |  |  | Control | Control | Control | Control | Control | Control |
| Faculty |  |  |  |  | Control | Control | Control | Control | Control |
| Only child or not |  |  |  |  |  | Control | Control | Control | Control |
| Student leader experiences |  |  |  |  |  |  | Control | Control | Control |
| Family socioeconomic position |  |  |  |  |  |  |  | Control | Control |
| Political status |  |  |  |  |  |  |  |  | Control |
| N | 777 | 777 | 777 | 777 | 777 | 777 | 777 | 777 | 777 |
| $R^2$ | 0.002 | 0.002 | 0.039 | 0.047 | 0.079 | 0.089 | 0.093 | 0.102 | 0.105 |
| Adjusted $R^2$ | 0.001 | 0.000 | −0.005 | 0.000 | 0.018 | 0.027 | 0.030 | 0.036 | 0.035 |

Note: * $p < 0.1$, ** $p < 0.05$, *** $p < 0.01$. The more asterisks, the higher the level of significance.

### 4.3.3. Regression Analysis of the Average Annual Time of Volunteer Service and the Self-Rating of Collective Responsibility

The regression results show that with the gradual addition of control variables, the goodness of fit of the model gradually increased, and after the addition of the control variable of whether the student was an only child or not, the average annual time of the model significantly affected collective responsibility. The one-by-one addition of control

variables such as student leader experience, family socioeconomic position, and political status did not increase or decrease such significance (Table 18).

**Table 18.** Regression results of average annual time to collective responsibility.

| | (1) | (2) | (3) | (4) | (5) | (6) | (7) | (8) | (9) |
|---|---|---|---|---|---|---|---|---|---|
| Average annual time | 0.002 (0.001) | 0.002 (0.001) | 0.001 (0.001) | 0.001 (0.001) | 0.002 (0.001) | 0.002 ** (0.001) | 0.002 ** (0.001) | 0.002 ** (0.001) | 0.002 ** (0.001) |
| Constant term | 4.112 *** (0.042) | 4.098 *** (0.103) | 4.057 *** (0.186) | 4.152 *** (0.193) | 3.984 *** (0.253) | 4.081 *** (0.264) | 4.145 *** (0.276) | 4.227 *** (0.340) | 4.291 *** (0.495) |
| Gender | | Control | Control | Control | Control | Control | Control | Control | Control |
| Province | | | Control | Control | Control | Control | Control | Control | Control |
| Grade | | | Control | Control | Control | Control | Control | Control | Control |
| Faculty | | | | Control | Control | Control | Control | Control | Control |
| Only child or not | | | | | | Control | Control | Control | Control |
| Student leader experiences | | | | | | | Control | Control | Control |
| Family socioeconomic position | | | | | | | | Control | Control |
| Political status | | | | | | | | | Control |
| N | 777 | 777 | 777 | 777 | 777 | 777 | 777 | 777 | 777 |
| $R^2$ | 0.003 | 0.004 | 0.044 | 0.049 | 0.096 | 0.098 | 0.099 | 0.105 | 0.108 |
| Adjusted $R^2$ | 0.002 | 0.001 | 0.001 | 0.002 | 0.037 | 0.038 | 0.037 | 0.039 | 0.039 |

Note: * $p < 0.1$, ** $p < 0.05$, *** $p < 0.01$. The more asterisks, the higher the level of significance.

## 5. Discussions

Education for sustainable development is a matter of great urgency. Many global issues urgently require the young generation to contribute their intelligence, and various carriers that enhance learners' ability to cope with the challenges of the era deserve attention. Participation in various voluntary services by university students during their studies is generally thought to help them better understand society and improve certain abilities. Internationally, many quantitative studies and longitudinal observations have been conducted on the value of volunteer service for local students. For Chinese students, however, these studies have often failed to perceive the true value of volunteer service on students' growth, as they have tended to prioritize political attributes. In fact, in the past three decades of rapid development in China, although volunteer services are of a national nature, students still have had a lot of autonomy in choosing which activities to participate in. Student volunteers are especially helpful for supporting the disabled, the elderly, and the children of migrant workers. These strengths make volunteer service an important bridge connecting the whole country and individual citizens, and a necessary channel for college students to enter the sustainable citizenship paradigm. Furthermore, the difficulty of obtaining data has also been a major obstacle for foreign scholars to conduct such studies. Lifting the veil on university student volunteers in China will give us a more thorough and deeper understanding of the hallmarks of volunteer service among Chinese university students.

For the first set of data in the study, objective data that could not be collected in the past, but can now be collected by a convenient information system, was adopted. We used the difference-in-differences method, which has mainly been used for policy effect assessment in the past. Natural comparative samples of special significance have increased the richness of similar studies. Our results show that even if students are stimulated to participate in voluntary service through external incentives, they still show substantial gains in terms of study and employment, which confirms that higher education institutions' behavioral incentives to students can promote volunteers' participation motivation [48]; this poses challenges to previous studies that have proposed that mandatory external incentives such as graduation requirements would hinder individuals' willingness to participate in activities [29]. Moreover, this study further clarifies the value of effective external incentives

to the growth and change of students who participate in volunteer service, which changed our simple assumption that volunteering is only beneficial with spontaneous participation.

Our analysis of the second set of data confirms the sustainability value of volunteer service, but also makes a distinction in judging the degree of its influence on the growth of various abilities for university students. Our study shows that the teamwork ability acquisition effect proves to be the strongest, showing the important value of volunteer service for the cultivation of collectivism for university students. This value possesses far-reaching significance, especially in China's current social system. Orderly group development is a strong follow-up to the concept of sustainability. The influence of natural characteristics on ability acquisition found through the study is similar to the cases found in Malaysia [49], providing a basis for providing more differentiated volunteer service programs to students with various backgrounds.

The third set of data was used to study the sense of responsibility as a separate object, and we distinguished between the senses of personal and collective responsibility. Research conclusions show that in specific samples of the study, students enhanced their sense of personal responsibility through volunteering, which supplements the common belief that volunteering can enhance university students' sense of social responsibility; this finding is worth noticing [50]. Therefore, it is possible that volunteering can enhance students' overall sense of social responsibility by satisfying their sense of personal responsibility and then satisfying their sense of collective responsibility; this calls for further in-depth study and discussion.

On the whole, the study provides a glimpse into the credible educational effects of the civic governance model with Chinese characteristics in the new era. We analyzed the growth of university students through volunteer service from an objective and realistic perspective without being affected by the political perspectives commonly pursued by foreign scholars who have observed volunteer service in China. The three sets of data in this study were examined from both objective and subjective perspectives, and the sustainability value of students' participation in volunteer service for personal growth was explored in a panoramic manner. Based on previous studies, more attention was paid to detailed discussions of the internal mechanisms of volunteer service influence. The adoption of new methods also brought to the study good ductility, which may provide enlightenment for other similar studies.

However, due to the limitations of sample collection, the data were restricted to one university. In the future, research can be extended to more universities, both in and outside China. Continuous data analysis on a larger scale and over a longer term is suggested to further prove the unique and sustainable value of volunteer service for university students all over the world.

## 6. Conclusions

Based on the data from a typical sample university, the paper first discussed the impact of volunteer service on university students' academic performance and employment, and on the likelihood of a sustainable future. Empirical results show that under the influence of relevant incentive policies, the more that students participate in volunteer service, the more significantly their academic performance and employment prospects will improve. Specifically, the difference-in-differences method was used, taking students from Faculty A who were affected by the scholarship policy as the treatment group, and other students who were not affected as the control group. Empirical analysis was conducted on the full sample data of students from classes from 2012 to 2015. Results show that students affected by the policy increased their volunteer service time by 70.613 h, average *GPA* by 0.033, and employment score by 0.343. This also shows that more volunteer service enables students to achieve better academic performance and employment.

Secondly, research and analysis of the expectations and results of ability acquisition in the sample university show that volunteer service is helpful for improving students' personal ability, teamwork ability, and leadership ability. There are significant differences

in the ability acquisition expectations of students in different universities, possibly due to the different environments or areas of expertise in different universities. Faculty, only-child status, political status, and student leader experiences all had an impact on students' actual ability acquisition, possibly because these four attributes are potential causes of their long-term psychological characteristics.

Finally, analysis was conducted on the impact of volunteer service on students' sense of responsibility, i.e., the increase in their self-ratings of personal responsibility, collective responsibility, and overall responsibility with the addition of average annual volunteer service time. Results show that for every one-hour increase in average annual volunteer service time, the scores of overall responsibility, personal responsibility, and collective responsibility increased by at most 0.003, 0.003, and 0.002 points, respectively. For every 100 h increase in average annual volunteer service time, the scores of overall responsibility, personal responsibility, and collective responsibility increased by 0.3, 0.3, and 0.2, respectively. This finding proves the importance of volunteer service to the cultivation of students' sense of social responsibility, and enriches the meaning students derive from participation in more volunteer service under relevant policy incentives.

**Author Contributions:** L.C.: contributed to the conception and design of the study, organized the database, performed the statistical analysis, and wrote the first draft of the manuscript. D.L.: validated the statistical analysis and revised sections of the manuscript. Y.L.: contributed to manuscript revision and read and validated the submitted version. All authors listed have made substantial, direct, equal, and intellectual contributions to the work and have approved it for publication. All authors have read and agreed to the published version of the manuscript.

**Funding:** This research was funded by the following: the 2019 Beijing Universities Ideological and Political Work Research Course Project (Research on Innovating the training system of volunteer service talents in Universities from the perspective of establishing "Four Correct Understandings"), grant number: BJSZ2019YB24; 2022 University-level Ideological and Political Project of the University of International Business and Economics (An empirical study on the value of Volunteer Service in College Labor Education—A case study of University of International Business and Economics), grant number JYX202203; and 2021 National Natural Science Foundation Youth Science Foundation Project (Research on Cooperation of Industry-University-Research Institute; the Optimization of Science and Technology Resource Allocation and Improvement of Innovation Performance), grant number 72104012.

**Institutional Review Board Statement:** This article does not contain any studies with animals, but involves survey and questionnaires on the self-reported ability of students. All participants of the survey were fully informed before filling in the questionnaire, and the data did not involve ethical issues such as their privacy.

**Informed Consent Statement:** Not applicable. It was considered by the members of the group.

**Data Availability Statement:** The labeled data set used to support the findings of this study is available from the corresponding author upon request.

**Conflicts of Interest:** The authors declare no conflict of interest.

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
