# Peer review of "Does Volunteer Service Foster Education for A Sustainable Future?—Empirical Evidence from Chinese University Students"

_sustainability, doi:10.3390/su151411259_

Round 1
Reviewer 1 Report
It is useful to investigate the impact of volunteer experience on Chinese university students' GPA, employability, and learning. This study provides an update on findings from a similar 2015 study.
There are some grammatical issues to resolve. For example, the title should be "Does Volunteer Service Fosters Education for a Sustainable Future?—An Empirical Evidence from Chinese University Students."
Author Response
Thank you very much for your valuable suggestions and comments. Please kindly find the following point-to-point responses:
Point 1: There are some grammatical issues to resolve. For example, the title should be "Does Volunteer Service Fosters Education for a Sustainable Future?—An Empirical Evidence from Chinese University Students."
Response 1: The title has been modified as “Does Volunteer Service Foster Education for A Sustainable Future? —An Empirical Evidence from Chinese University Students”. Besides, we have made numerous corrections and changes throughout the paper to correct the grammar mistakes and language errors. References are modified and up-to-dated.
Reviewer 2 Report
The article is interesting and addresses a topic relevant for higher education today. It responds, to a large extent, to the rationale of the special issue. Methodologically it is innovative and the results are presented in a manner that stirs curiosity and fuels the debate on the role of service learning (volunteering) during student years. Having said that, I believe the authors need to dedicate more effort to increase the depth of their analysis. My main concerns are the following:
1. The literature review is insufficiently developed. Volunteering deserved more attention and student volunteering is also more broadly discussed in scientific literature.
2. The link between volunteering and sustainability, present in the title, needs a stronger unfolding.
3. The presentation of results is interesting, but the discussion needs more unfolding, both in terms of the work carried out for this article, and in the context of the literature on student volunteering (in China and worldwide).
4. The Conclusions should be presented separately from the Discussion section.
I encourage the authors to do the work and improve the overall presentation of their results.
English language editing is recommended, starting from the title. Correctly is should read: Does Volunteer Service Foster Education for Sustainable Future?—An Empirical Evidence from Chinese University Students
Author Response
Thank you very much for your valuable suggestions and comments. We have made numerous corrections and changes throughout the paper to correct the grammar mistakes and language errors. Please kindly find the following point-to-point responses:
Point 1: The literature review is insufficiently developed. Volunteering deserved more attention and student volunteering is also more broadly discussed in scientific literature.
Response 1: In this new version, 14 literatures are added to track the latest research on college students' volunteer service. References are modified and up-to-dated.
Point 2: The link between volunteering and sustainability, present in the title, needs a stronger unfolding
Response 2: The new version rewrites the first paragraph of the introduction, adding evidence of a strong link between sustainability and volunteerism, logically clarifying the relationship between the two.
Point 3: The presentation of results is interesting, but the discussion needs more unfolding, both in terms of the work carried out for this article, and in the context of the literature on student volunteering (in China and worldwide).
Response 3: We re-extended the cognitive background of this study to a certain extent, supplemented the use process and research enlightenment of the three sets of data in this paper, and elaborated in literature review, discussion and other parts.
Point 4: The Conclusions should be presented separately from the Discussion section.
Response 4: Thank you very much for your suggestion. The new version has separated the discussion and conclusion as requested, and discussed the relationship between volunteerism and ESD through a simplified but strong conclusion.
Reviewer 3 Report
The authors use The difference-in-differences method that was introduced into the field of economic re- 162 search in the 1970s [22] in order to study the influence of th volonteering work of students on their Education for Sustainable Future. Below are some comments:
1- The authors say: This section uses the hierarchical regression model and the expression is as follows: 393 ???? = ?0 + ?????? + ???(i = 1,2 … … n) (4), i represents different students, and ???? refers to the score of overall responsibility. Well generally, the regression is done on the average of the variable and not on each students' score. The authors need to explain what they mean.
2- Explaining the above formula, the authors say: "?? refers to a series of control variables, including gender, family province, grade, college, only child or not, student leader experience, family socioeconomic position, and political status. " How could Xi refer to a series when it represents a variable?
3- The authors say: "Results and Findings". What is the difference between the two?
4- Why isn't there mention for some variables, like employment, in 3.3.2. The Analysis Method?
5- Why don't the authors use the agreed form of presenting the results of regression analysis? They should say that.
6- The discussion section is very short and does not cover all the issues presented in the results section.
7- The authors should divide the discussion and conclusions section into two sections: discussion section and conclusions section.
8- The discussion section should integrate sufficient previous studies.
Acceptable
Author Response
Thank you very much for your valuable suggestions and comments. We have made numerous corrections and changes throughout the paper to correct the grammar mistakes and language errors. 14 new references are added, modified and up-to-dated. Please kindly find the following point-to-point responses:
Point 1: The authors say: This section uses the hierarchical regression model and the expression is as follows: ???? = ?0 + ?????? + ??(i = 1,2 … … n) (4), i represents different students, and ???? refers to the score of overall responsibility. Well generally, the regression is done on the average of the variable and not on each students' score. The authors need to explain what they mean.
Response 1: Thank you very much for pointing out the problem. The original article used an irregular vector representation, and the new version has adjusted the form of the equation. (4)
In this model, the mean value of students' comprehensive responsibility score is returned, and the change of goodness of fit of the model is observed by adding control variables one by one, indicating that the inclusion of control variables is conducive to the fitting of the model.
Point 2: Explaining the above formula, the authors say: "?? refers to a series of control variables, including gender, family province, grade, college, only child or not, student leader experience, family socioeconomic position, and political status. " How could Xi refer to a series when it represents a variable?
Response 2: Thank you very much for pointing out the problem. The original article used an irregular vector representation, and the new version has adjusted the form of the equation. For instance,
(4)
In which represents the comprehensive sense of responsibility score, represents the annual average volunteer time, and to represent a series of control variables, including gender, family province, grade, college, whether the only child, whether being the former student leader, family economic status, and political status. Control variables are added one by one.
Point 3: The authors say: "Results and Findings". What is the difference between the two?
Response 3: Thank you for your suggestion. The title has been changed to “Results”.
Point 4: Why isn't there mention for some variables, like employment, in 3.3.2. The Analysis Method?
Response 4: This part mainly studies the relationship between annual average volunteer time and sense of responsibility, for which the relevant situation of employment is not used. The study focuses on the impact of volunteer service participation on students’ academic achievement and employment as dependent variables in 3.1.2.
Point 5: Why don't the authors use the agreed form of presenting the results of regression analysis? They should say that.
Response 5: Traditional linear regression can output the coefficient and significance of each independent variable and control variable, so as to judge the degree of their influence on the dependent variable. However, the questionnaire data used in this paper makes a control variable composed of multiple dummy variables, so that linear regression will output multiple coefficients, which is difficult to reflect the effect of the whole set of control variables. Therefore, the hierarchical regression method is adopted to reflect the influence of control variables through the change of goodness of fit before and after the addition of the whole group of control variables.
The reason why the coefficients of all variables are not marked as in the standard form is that, on the one hand, as dummy variables, the size of the coefficients is not very helpful to the interpretation of the equation results; on the other hand, the tens of coefficients all marked will make the whole table too complicated and not concise enough.
Point 6: The discussion section is very short and does not cover all the issues presented in the results section.
Response 6: In the revised edition, we have enhanced the discussion section to include and correspond to the content of the conclusion section.
Point 7: The authors should divide the discussion and conclusions section into two sections: discussion section and conclusions section.
Response 7: Thank you very much for your suggestion. The new version has separated the discussion and conclusion as requested, and discussed the relationship between volunteerism and ESD through a simplified but strong conclusion.
Point 8: The discussion section should integrate sufficient previous studies.
Response 8: Your advice is of great value to us. The new edition has added relevant studies in addition to those already mentioned in the literature review to the discussion section to enrich the content and integrate more previous studies.
Round 2
Reviewer 2 Report
I see that the remarks on the first version of the paper were addressed. I believe the article gained clarity and contributes to scholarship in student volunteering.
Minor editing will be needed.
Reviewer 3 Report
Thanks
O.K.